Identification of potential inhibitors of the main protease from feline infectious peritonitis virus using molecular docking and dynamic simulation approaches

Khan Mohd Yasir 1
Shah Abid Ullah 2
Duraisamy Nithyadevi 1
Moawad Nadine 2
ElAlaoui Reda Nacif 1
Cherkaoui Mohammed 1
Hemida Maged Gomaa maged.hemida@liu.edu 2
1 Department of Computer Science, College of Digital Engineering and Artificial Intelligence, Long Island University , Brooklyn , NY , United States of America
2 Department of Veterinary Biomedical Sciences, College of Veterinary Medicine, Long Island University , Brookville , NY , United States of America
García-Contreras Rodolfo
Electronic publication date: 2025 Jul 30
Publication date: 2025
Volume: 13
Electronic Location ID: e19744
Received 2025 Apr 14; Accepted 2025 Jun 23
Copyright: ©2025 Khan et al.
Copyright year: 2025
Copyright holder: Khan et al.
License: This is an open access article distributed under the terms of the Creative Commons Attribution License, which permits unrestricted use, distribution, reproduction and adaptation in any medium and for any purpose provided that it is properly attributed. For attribution, the original author(s), title, publication source (PeerJ) and either DOI or URL of the article must be cited.
License URL: https://creativecommons.org/licenses/by/4.0/

Keywords: FIPV, Mpro, In silico design, Molecular docking, Antiviral, Dynamic simulation, Nucleoside analogs, GS441524, Sofosbuvir, FCoV

Funding: Long Island University 36524 This study was funded by a seed grant from Long Island University (Grant no: 36524). The funders had no role in study design, data collection and analysis, decision to publish, or preparation of the manuscript.

==============================
Background

Feline infectious peritonitis virus (FIPV) is one of cats’ most serious viral infections. The FIPV infection induces a complicated syndrome in the affected cats, including immunosuppression and severe inflammatory conditions. Unfortunately, vaccines are unable to provide complete prevention in cats from getting infected with these viral infections. There is ongoing research on preparing antiviral therapies against FIPV in cats. However, these are still in clinical trials and have not been fully approved by the drug authorities in many countries, including the USA. Targeting the main viral proteases is one of the promising trends in the drug design of many viral diseases, including coronaviruses. The main goal of the current study was to repurpose and test the efficacy of some known antiviral drugs to treat FIPV infection in cats by targeting the FIPV main protease (Mpro).

Methods

We used the in-silico prediction and molecular docking tools to screen and identify some drugs targeting FIPV-MPro to achieve these goals. The research method was started by building a screening pharmacokinetic associated variables of the compound, then used to design a new potential inhibitor by employing the docking and molecular dynamic simulation to evaluate the interaction of all complexes using the standard dynamics cascade protocol of Biovia Discovery studio.

Results

Our results show that out of the 15 antiviral and immunomodulatory compounds, the top-ranked inhibitors for the FIPV-Mpro are reference standard inhibitor (N3), Sofosbuvir, and the GS-441524, out of which GS-441524 was suggested as Mpro-inhibitor on the basis of further investigation through molecular dynamics simulation method. In conclusion, our results confirmed the potential applications of the predicted FIPV-Mpro inhibitors either independently or in combination with other immune-modulatory compounds. Further in vitro and in vivo studies are encouraged to test the efficacy of these identified compounds as potent inhibitors for the Mpro of the FIPV in cats. This study will pave the way for the development of novel drugs that treat FIPV infection in cats.

Introduction

Feline infectious peritonitis virus (FIPV) is one of the leading causes of death in naïve cats under three years of age (Thayer et al., 2022). However, some recent reports show older cats more than three years old could get severe FIPV infection, which may lead to the death of the infected animals (Shah et al., 2025). The FIPV infection in cats is usually associated with various clinical signs. Some cats show non-specific clinical signs such as inappetence, lethargy, and jaundice. Some cats develop a wet/effusive form, which is mainly characterized by asities, pale color of the mucus membranes, dyspnea, and pleural effusion (Felten & Hartmann, 2019). The dry form could also seen in some FIPV-infected cats in which the affected animal shows some neurological disorders such as seizer. Some ocular pathology could also seen in the cats infected with the dry form of FIPV, such as uveitis, keratitis, and retinal displacement (Felten & Hartmann, 2019).

Diagnosis of FIPV in cats is very tricky, and there are no gold standard clinical signs, techniques/assays for the diagnosis of FIPV in cats (Thayer et al., 2022). However, if some cats are presented with some effusive forms, testing aspirate of this fluid showed higher sensitivity in the confirmation of FIPV infection compared to other blood tests (Felten & Hartmann, 2019). The clinical pathological examination of various body fluids of the infected animals, including the blood, urine, sera, and effusion fluid, could assist with the FIPV diagnosis. The FIPV-infected cats are most likely to develop some types of anemia, lymphopenia, hyperbilirubinemia, hyperglobulinemia (Riemer et al., 2016). Diagnostic imaging play an important role in the diagnosis of FIPV and in the monitoring of the progression of the disease as well as in the success of the application of some antiviral compounds against FIPV infection in cats (Schreurs et al., 2008).

The histopathology and immunohistochemistry using some biopsy and tissue species from freshly dead cats would be an asset in the diagnosis of the FIPV infection (Stranieri et al., 2020). One of the highly effective approaches for FIPV diagnosis is the detection of the viral antigen in the tissues of infected animals with the immunohistochemistry (IHC), in association with the detection of the viral protein in the macrophages (Stranieri et al., 2020). Detection of antibodies in sera of cats against the feline coronavirus (FCoV) did not support the diagnosis of FIPV in cats due to the shared antigenic ship between those two forms of the virus (Hartmann et al., 2003). Application of the reverse transcription polymerase chain reaction (RT-PCR) using primers designed against various viral proteins, including the M, N, and S, of limited importance in the diagnosis of FIPV except the assay that could detect the mutated version of the FCoV that trigger the FIPV infection in cats (Meli et al., 2004).

There are several challenges associated with the design and development of FIPV drugs, including (1) the complex nature of the FIPV infection in cats, (2) the lack of the gold standard diagnostic assays/markers for the viral infection, (3) the high cost of the newly designed drugs, (4) the large number of animals require to validate the efficacy of some new drugs, the possibility of relapse of the treats cats hampered the design and success of the FIPV drugs. All these factors hampered the success of the newly developed FIPV drugs in cats. There is a high demand for the design, development, and validation of some novel drugs for the treatment of FIPV in cats, particularly in shelters, cats in crowded populations, and cats in the same household or catteries, in addition to immunocompromised cats.

FIPV belongs to the order Mononegavirales, family coronaviruses and genus alphacoronavirus, species alphacoronavirus-1 and subspecies feline coronavirus (FCoV).1 The viral genome is a single molecule of a positive sense RNA. The FIPV genome has a typical coronavirus genome organization and encodes 11 proteins, including four structural and seven non-structural proteins. The major non-structural proteins encoded by Gene-1 (ORF1a and ORF1b with ribosomal frameshifting in between) reside at the 5′ two-thirds of the genome. Other non-structural proteins of the FCoV are encoded by the three ABC and 7/A/B genes, while the structural proteins (Spike (s), Envelope (E) Membrane (M), and the nucleocapsid (N)). The ORF1A/B is further processed by some virally encoded proteases into 16 non-structural proteins (NSP-1-NSP16) (Brierley, Digard & Inglis, 1989). The FIPV-3c protein is important for virus replication and contributes to viral virulence and tissue tropism (Jaimes & Whittaker, 2018). Although the function of the FIPV-7ab proteins is not fully understood, they might play important roles in viral immune evasion, particularly as an antagonist for the IFN-type I (Dedeurwaerder et al., 2013).

Based on the FIPV-S sequences, the virus is classified into two serotypes. The serotype-I is designated as FIPV, which is the most virulent serotype and causes a lethal infection in cats (Hohdatsu, Okada & Koyama, 1991). Although FIPV infection in cats is not contagious (it does not spread easily among cat populations) until now, the prognosis of the cats infected with the virus is always fatal in most cases (Kennedy, 2020). The absence of vaccines that could protect cats against FIPV infection makes antiviral therapy the only remedy to treat cats from FIPV infection. Several antiviral compounds, particularly nucleoside analogs and interferons, have been tried to inhibit or interrupt the FIPV replication cycle at various stages (Addie et al., 2020; Dickinson et al., 2020; Murphy et al., 2018; Pedersen et al., 2019). This approach includes the application of a single or a combination of different compounds together to ensure the robust inhibition of viral replication (Schmied et al., 2024).

The main protease (Mpro) is a potent enzyme in the life cycle of coronaviruses and plays a pivotal role in viral replication and transcription. It is responsible for proteolytically cleaving the large viral polyproteins conserved sites to release functional proteins required for the formation of the replication-transcription complex. Structurally, the Mpro protein consists of three domains and contains a catalytic dyad composed of His41 and Cys145, which are highly conserved among coronaviruses. Its crucial function in viral replication makes the Mpro an ideal target for antiviral drug development, as inhibition of its activity can effectively block viral replication without interfering with host cellular processes. Therefore, coronaviruses Mpro is one of the most important targets for the design of antiviral therapy for several coronaviruses, including the Middle East respiratory syndrome coronavirus (MERS-CoV) and the severe acute-2 (SARS-CoV-2), the porcine epidemic diarrhea virus (PEDV) as well as the FIPV (Wang et al., 2020).

The main rationale of this study is based on the inhibition of some key viral proteins that play essential roles in viral replication, particularly in the processing of the viral polyprotein, which proved to have strong inhibitory effects on the replication of many coronaviruses, including the FIPV. The major objective of the current study is to use virtual screening of a large number of antiviral compounds that most likely inhibit the FIPV-Mpro expression and, thus, inhibit viral replication.

The main reason for targeting CoVs-Mpro is their essential role in polyprotein processing during viral replications. In the current study, we used the in-silico drug design tools to predict the efficacy of 15 selected antiviral, anti-inflammatory, and immune-modulatory compounds on the inhibition of the FIPV-Mpro enzyme (standard inhibitor-N3) compound and nucleoside precursors/analog such as oxipurinol, favipiravir, pentoxifylline, baricitinib, methotrexate, gemcitabine, galidesivir, ribavirin, 6-Azauridine, GS441524, mizoribine, sofosbuvir, molnupiravir, and tenofovir. We used the crystal structure of the FIPV-Mpro available in the public domain and analyzed the binding sites of each compound to these in the FIPV-Mpro.

Although some other studies used molecular docking approaches to screen and design some other antiviral therapies for FIPV infection in cats, most of these studies have focused on limited compound libraries or lacked comprehensive validation through molecular dynamics simulations. The major novelty of this study is the integrative of the in silico approach, combining pharmacokinetic profiling, molecular docking, MM-GBSA free energy calculations, and extended molecular dynamics simulations to evaluate the inhibitory potential of a broad panel of nucleoside analogs against the FIPV main protease (Mpro). The inclusion of clinically relevant compounds such as GS-441524 and Sofosbuvir, along with rigorous computational validation, make this study more unique than the earlier studies (Theerawatanasirikul et al., 2020).

The research approach in this study will provide more insights into the inhibitory actions of the selected compounds on the FIPV-Mpro. These potential FIPV drugs could be used independently or in combination to inhibit FIPV replication, suppress the severe inflammatory conditions associated with viral infections, and enhance the immune response against the viral infection. It will also pave the way for more research to do more functional characterization of the potential inhibitory effects of these compounds on the FIPV replication in the in vitro and in vivo models. This approach will lead to the production of effective antiviral therapy against FIPV infection in cats.

Materials & Methods

Receptor protein and ligand file preparation

The main protease (FIPV-Mpro) protein in complex with the ligand N-(5-methylisoxazol-3-yl)-3-tosylpropanamide (N3), and its 3D structure was downloaded from the RCSB-Protein Data Bank (PDB) (https://www.rcsb.org/), in the PDB file (Ali et al., 2017). The protein data bank is a library for biological compounds that stores three-dimensional structural information. The PDB ID of FIPV N3-Mpro complex is 5EU8. Further, all nucleoside precursors and analogs as ligands were retrieved from PubChem database (https://pubchem.ncbi.nlm.nih.gov). The PubChem compound ID (CID) for each ligand is provided in Table 1. The protein was served as receptors in the docking process. The files were opened using BIOVIA Discovery Studio Visualizer 2024. Water molecules and ligands attached to the receptor were removed, and the receptor protein stored in the pdb format. Further, using prepare-protein and prepare-ligand protocol of Biovia Discovery Studio v24.1.0.321712, polar hydrogen atoms were added to the receptor and ligands. Subsequently, the protein and ligand files file were saved in the PDB format.

Table 1 Summary of the chemical properties of selected nucleoside precursors and analogs.

S.N.	Compound name	Pubchem iD
(CID)	M.WT	ALogP	HBA	HBD	LPV	MPSA (Å2)	
1	6-Azauridine	5901	245.189	−2.432	9	4	0	131.69	
2	Baricitinib	44205240	371.417	0.36	9	1	0	128.94	
3	Favipiravir	492405	157.103	−1.392	5	2	0	84.55	
4	Gemcitabine	60750	263.198	−1.394	7	3	0	108.37	
5	GS-441524	44468216	291.263	−1.43	9	4	0	149.91	
6	Molnupiravir	145996610	329.306	−0.875	10	4	0	140.91	
7	Oxipurinol	135398752	152.111	−0.867	6	3	0	86.88	
8	Pentoxifylline	4740	278.307	0.507	7	0	0	75.51	
9	Ribavirin	37542	244.205	−2.745	9	4	0	143.72	
10	Tenofovir	464205	287.212	−0.772	9	3	0	146.19	
11	Galidesivir	10445549	265.269	−1.582	8	7	1	140.31	
12	Methotrexate	126941	454.439	0.114	13	7	2	210.53	
13	Mizoribine	104762	259.216	−2.409	9	6	1	151.06	
14	Sofosbuvir	45375808	529.453	0.921	12	3	2	152.54	
15	*N3	146025593	680.791	3.54	14	6	3	204.74	
Notes.

* The reference standard inhibitor of FIPV-Mpro activity.

M.WT Molecular Weight

HBD Hydrogen bond doner

HBA Hydrogen bond acceptors

MPSA Molecular polar surface area

LPV Lipinski Violation

ADMET and drug-likeness analysis of selected ligands

The selection of nucleoside precursor and analog compounds used as ligands in this study was based on previous reports on the antiviral activity of these compounds (Borbone et al., 2021; Zenchenko, Drenichev & Mikhailov, 2021). The selected nucleoside precursors and analogs were screened for detailed analysis of physicochemical descriptors, drug-likeness through Lipinski rule of five and pharmacokinetics-associated variable i.e., absorption, distribution, metabolism, and excretion and toxicity (ADMET) (Bultum et al., 2022). In order to assist the early identification of potential mutagens, in silico prediction of compound mutagenicity, the Ames test is the most widely used assays for testing the mutagenicity assisted toxicity of a compound (Chu et al., 2021). The AMES toxicity prediction and ADMET properties were analyzed by using the AMES Test and ADMET protocol of Biovia Discovery Studio v24.1.0.321712.

Redocking-validation of target protein-ligand interaction

For the validation of docking of ligands to the target protein, the Mpro receptor (PDB ID:5EU8) was docked with its cocrystallized native ligand inhibitor N3 N-[(5- methylisoxazol-3-yl) carbonyl] alanyl-L-valyl-N∼1∼-((1R, 2Z)-4-(benzyloxy)-4-oxo-1-{[(3R)-2-oxopyrrolidin-3-yl] methyl} but-2-enyl)-L-leucinamide. The method is said to be valid if the RMSD value obtained is ≤2°A, so that docking of the test compound can be carried out with the target protein in the same radius of sphere site.

Docking of compounds with Mpro

Mpro helps in the replication of the viruses and thus becomes an important antiviral drug target. Protease inhibitors are effective in blocking the coronavirus replication and proliferation by interfering with the post-translational processing of essential viral polypeptides. The crystal structure of Mpro, especially the co-crystallized form with the inhibitor N3 (PDB ID: 5EU8), has been widely used for structure-based drug discovery efforts, including molecular docking and virtual screening. The molecular docking approach can be used to model the interaction between a Mpro protein with nucleoside precursors and nucleoside analogs. The interactions between these small molecules and proteins at the atomic level, molecular docking helps us understand how these molecules bind and function within biological processes (Agu et al., 2023; Meng et al., 2011). In the CDOCKER tool from Biovia Discovery Studio docking protocol, we typically use 10 docking poses for ligand. For each dock ligand pose, the higher positive values of CDOCKER interaction energy score and calculated binding energy (ΔG) indicates more favorable binding (Agu et al., 2023; Ding et al., 2020; Gagnon, Law & Brooks 3rd, 2016).

These computational tools enable the visualization of the ligand-target interaction (molecular docking) and the identification of the compounds that bind more efficiently with the target (Agu et al., 2023). This analysis typically involves examining the docking scores, ligand-protein interactions, and the visualization of the docked complexes. To describe the defining binding site in protein, interaction binding affinity score (-CDOCKER interaction score), calculation of binding energy (ΔG), is likely an internal step within Biovia Discovery Studio v24.1.0.321712.

MM-GBSA calculations—molecular mechanics-generalized born surface area

The binding free energy of the observed protein-ligand complexes was measured using the MM-GBSA approach, which integrates molecular mechanics (MM) force fields with a generalized Born and surface area continuum with none-implicit solvation model. The MM-GBSA calculation incorporates the CHARMm force field, with partial charge estimation using Memory Rone. The Di-electric constant is 1 and the minimum non-bond higher and lower cuttoff distance is 12 and 10 Å. The MM-GBSA was determined in this study using the equation ΔG = E_complex (minimized)–(E_ligand (minimized) + E_receptor (minimized)) from Biovia Discovery Studio v24.1.0.321712. The default setting employed to compute MM-GBSA involved rendering all protein atoms rigid while the ligand atoms are relaxed.

Molecular dynamics simulation

Based on in vitro analysis, docking energies, and conformational pose analysis, the four best complexes were selected for molecular dynamics (MD) simulation (Ahmad et al., 2021; Arnittali, Rissanou & Harmandaris, 2019). To further explore the dynamic behavior of ligand-protein complexes, the best predicted top hits of compounds and N3 (reference standard) with respect to Mpro-protein were selected to perform 50,000 picoseconds (ps) through standard dynamics cascade simulation. To generate the molecular topology files for protein complex and to create the topology of ligands, the CHARMm36 force field. The simulation system consists of an explicit boundary solvent model, orthorhombic box with a minimal distance of seven nm between the protein surface and edge of the box neutralized with the inclusion of cation type sodium (Na) and anion type chloride (Cl) counter ions. For the energy minimization, the steepest descent (minimization 1) with RMS gradient 1 and conjugate gradient (minimization 2) with RMS gradient 0.1. Both minimization algorithms were used for 50,00 steps. The heating phase was performed using simulation time four ps with time step two fs; for immersion the initial temperature is 50 and target temperature 300 K with save results interval two ps.

Equilibration phases were carried out for 200 ps (picosecond) simulation run with two fs (femtosecond) time step and the save result interval is two ps. The Particle–Mesh–Ewald (PME) algorithm was used for long-range electrostatic interactions with fourth-order cubic interpolation and kappa 0.34 Å grid spacing. The advance dynamic integrator used Leapfrog Verlet algorithm with apply shake constraint. The implicit solvent model was used with dielectric constant 1, Nonbond list radius cutoff 14 in which nonbond higher cutoff distance is 12 and nonbond lower cutoff distance is 10. The production step of a standard dynamic cascade of MD simulation was carried out for 50 ns (50,000 picoseconds). Trajectory analysis was done to confirm hydrogen bond distance, root mean square deviation (RMSD), root mean square fluctuation (RMSF), and radius of gyration (RG) of each system. The stability of the complex is indicated by the highest potential inhibitor from the stable complex protein-ligand through Biovia Discovery Studio v24.1.0.321712.

Results

Pharmacological potential of nucleoside precursors and analogs

For drug screening, Lipinski’s rule of five includes criteria such as molecular weight (M.W. ≤ 500 Da), hydrogen bond donors (HBD ≤ 5), hydrogen bond acceptors (HBA ≤ 10), and the octanol-water partition coefficient (LogP ≤ 5) (Ahmad et al., 2024). Our drug-likeness analysis portrayed that selected nucleoside analogs fall under the acceptable scores of Lipinski’s rules of five with a few exceptions. In particular, the chemical structure of methotrexate showed seven HBD and 13 HBA violated two rules of Lipinski’s, whereas compound sofosbuvir also violated two rules of Lipinski’s exhibited 12 HBA and M.W. 529, which we overruled for further screening. In contrast, the N3 (reference standard) exhibited 14 HBA, six HBD and M.W. 680.79 Da, violating rules of Lipinski’s (Table 1). The ALogP analysis revealed that all the nucleoside analogs with the standard N3 have LogP values below 5, which is within the desirable threshold (LogP ≤ 5). Compounds with a molecular polar surface area (MPSA) ≥ 140 Å2, struggle to cross cell membranes effectively, which could result in poor absorption and bioavailability, especially when taken orally. Compounds with an MPSA value between 75–140 Å2 are typically considered to have good permeability properties, while compounds with values higher than 140 Å2 are more likely to face issues with membrane permeability and absorption (Veber et al., 2002). In our study, all the compounds exhibited the range of MPSA below the desired threshold (140 Å2) except standard reference N3, methotrexate and sofosbuvir (Table 1).

ADMET variables and AMES test analysis of Nucleoside analogs and precursors

The parameter ADMET solubility is defined by the base 10 logarithm of the molar solubility in water (Log-SW), as predicted by the regression (Table S1). On the basis of Log (SW), we defined the solubility of compounds by categorical solubility level. The solubility analysis revealed that selected nucleoside analogs exhibited better aqueous solubility, ranging solubility level from 2–4 (Table 2), which shows its drug likeliness solubility level (Level 0: extremely low, 1: No-very low but possible, 2: Yes-but low, 3: Yes-good, 4: Yes-optimal, 5: No-too soluble). The good solubility compounds could be explained by the higher number of HBs formed by the nucleoside analogs in aquas solutions (Table 2 and Table S1). In ADMET method of Biovia discovery studio, blood brain barrier (BBB) model contains a quantitative linear regression model for the prediction of blood–brain penetration, as well as 95% and 99% confidence ellipses in the ADMET_PSA_2D, ADMET_AlogP98 plane. The model was derived from over 800 compounds that are known to enter the CNS after oral administration (Egan & Lauri, 2002). The model includes four prediction levels within the 95% and 99% confidence ellipsoids: level 0 (Very high penetrant), 1 (High), 2 (Medium), 3 (Low) and 4 (Undefined-No prediction is made for compounds outside the 95% and 99% confidence ellipsoids). BBB penetration level for the selected compounds ranged from 3 to 4, indicating either low or very low penetration and absorption across the blood–brain barrier (Table 2). Additionally, the estimation of the level of plasma protein binding (PPB) plays a crucial role in the distribution of a drug from circulation to the target organs. The plasma protein binding model predicts whether a compound is likely to be highly bound (≥ 90% bound) to carrier proteins in the blood. The negative values (ranging from −4.1131 to −38.845) indicate that these compounds do not have strong PPB (Table S1). These values likely reflect the binding affinity of the compounds to plasma proteins like albumin. More negative values typically suggest weaker binding or less affinity for plasma proteins. A low or weak binding indicates that a greater proportion of the drug will be available in its free (active) form in circulation.

Table 2 Summary of the predicted ADME properties and toxicity of selected nucleoside precursors and analogs.

S.N.	Compounds	Solubility level	BBB level	CYP2D6 prediction	Hepatoxicity prediction	Absorption level	PPB
prediction	TOPTAK AMES prediction	
1	*N3	3	4	false	false	3	false	Non-Mutagen	
2	6-Azauridine	5	4	false	true	3	false	Mutagen	
3	Baricitinib	3	4	false	true	0	false	Non-Mutagen	
4	favipiravir	4	4	false	true	1	false	Non-Mutagen	
5	Galidesivir	4	4	false	true	2	false	Mutagen	
6	Gemcitabine	4	4	false	true	1	false	Non-Mutagen	
7	GS-441524	4	4	false	true	2	false	Non-Mutagen	
8	Methotrexate	2	4	false	true	3	false	Non-Mutagen	
9	Mizoribine	5	4	false	true	3	false	Non-Mutagen	
10	Molnupiravir	4	4	false	true	2	false	Non-Mutagen	
11	Oxipurinol	4	3	false	true	0	false	Non-Mutagen	
12	Pentoxifylline	3	3	false	false	0	false	Non-Mutagen	
13	Ribavirin	5	4	false	true	3	false	Non-Mutagen	
14	Sofosbuvir	3	4	false	true	3	false	Non-Mutagen	
15	Tenofovir	4	4	false	true	1	false	Mutagen	
Notes.

* The reference standard inhibitor of FIPV-Mpro.

BBB blood brain barrier

PPB plasma protein binding

The key to intestinal absorption (IA), ADMET absorption level, is determined by the Mahalanobis distance (ADMET_Absorption_T2_2D) of the compound in the ADMET_PSA_2D and ADMET_AlogP98 plane. This distance is compared to the center of the region of chemical space defined by well-absorbed compounds. Based on this distance, the compound is categorized into one of four absorption levels (good:0, moderate:1, low:2, very low or poor:3). Therefore, the efficiency of drug relies on their intestinal absorption (IA) to the distribution to target organs where the absorption range are considered low, middle and high, respectively (Ahmad et al., 2021). The predicted absorption levels from ADMET model of all the compounds ranged from 0 to 3 (Table 2). Compounds with absorption levels of 0–1 are predicted to have good to moderate absorption, suggesting enhanced bioavailability, which may be linked to the potent pharmacological actions of these nucleoside precursors and analogs. In contrast, the reference Mpro-standard inhibitor (N3) and inhibitors such as methotrexate, sofosbuvir, mizoribine, ribavirin, and 6-Azauridine were predicted to have an absorption level of 3, suggested poor intestinal absorption for these inhibitors. In ADMET, CYP2D Prediction indicates the predicted classification, using a cutoff Bayesian score of 0.161 to minimize false positives and false negatives and the Bayesian score calculated by a model, used to classify a compound as either a CYP2D6 inhibitor or non-inhibitor (Kato, 2020). The CYP2D6 values for all the compounds are predominantly negative, ranging from −0.67 to −12.21 (Table S1). According to the ADMET CYP2D6 model, more negative values indicate weaker binding or less interaction with the enzyme. The model’s false predictions suggest that these compounds are not significant inhibitors or binders of the CYP2D6 enzyme (Table 2).

The hepatotoxicity prediction values represent the prediction scores or risk scores for hepatotoxicity. Typically, the score indicates the likelihood of hepatotoxicity, with negative scores often associated with a lower risk and positive scores associated with a higher risk (Table S1). The ADMET hepatotoxicity prediction values reflect the model’s output, where compounds with higher positive values (3.3959 and 23.9597) are more likely to be predicted as hepatotoxic (True), and those with negative values such as standard inhibitor N3 and pentoxifylline (−7.21011, −12.2658) are less likely to be hepatotoxic (False) (Table 2 and Table S1). On the other hand, the test for prediction (TOPTAK prediction methods) of mutagen detection is based on the Ames prediction. The Ames test values are mostly negative (Table S1), which indicates that these compounds are unlikely to cause mutations and do not pose a significant genetic risk in terms of mutagenicity. Compounds with positive or near-zero Ames values tend to be classified as mutagenic (Table 2).

Mpro-N3 redocking validation of the molecular docking process

To validate the further docking results of selected ligands, the FIPV-Mpro receptor protein was re-docked with N3. The redocking’s binding site area was x: −47.488, y: −14.895, and z: −9.396, with site sphere radius of 13.53. The parameter of the validation method is RMSD. The RMSD analysis showed the degree of deviation from experimental ligand docking results to the crystallographic ligand attached with FIPV-Mpro protein at the same binding site. The higher the RMSD value, the greater the deviation, which indicates the higher prediction error of ligand-protein interactions. Conversely, the low RMSD value is attributed to better conformation because the redocking ligand position is closer to the ligand position resulting from crystallography. The redocking results indicated a 1.0683 Å RMSD value from the native ligand with the Mpro receptor (Fig. 1). Therefore, based on the low RMSD value between cocrystallized native and re-docked ligand, it can be said that the method used for redocking in this study is valid and can be used against tested ligands with the same binding site area.

Figure 1 Results of redocking of the N3 compound with the co-crystallized native N3-Mpro complex.

The superimposition of the ligand position based on the redocking process of the N3 compound with the co-crystallized native N3-Mpro complex (cyan blue: crystallography; green: redocking).

Mpro-Ligands interaction and binding energy

MPro in complex with N3 compound in X-ray crystal structure from RCSB PDB database (PDB ID:5eu8; 1.80 Å resolution) was chosen as the receptor for different ligand docking. The PubChem CID for all nucleoside analogs as ligands is given in Table 1. In this attempt to filter suitable compounds with the best binding to FIPV-Mpro, our molecular docking studies revealed a strong interaction of nearly all the selected compounds with binding energies (ΔG) from −7.4 to −4.8 kcal/mol and CDOCKER interaction energy score for binding affinities from 68 to 14 against the active pocket of FIPV-Mpro (Table 3). Among nucleoside precursors and analogs, nucleoside analog Sofosbuvir topped the binding energy and affinity score (ΔG: −7.38 kcal/mol and CDOCKER interaction energy score: 39.34) and interacted with His41, Cys144, Leu164, Glu165, Leu166, Pro188 residues of FIPV-Mpro active pocket (Fig. 2 and Table 3). The interaction of another top-scoring nucleoside analog, i.e., GS-441524 with FIPV-Mpro, was favored by ΔG of −7.2 kcal/mol, whereas binding affinity score (CDOCKER interaction energy score: 43) and interacting residues of Mpro with GS-441524 is given in Table 3. The interaction pattern of the top two ligands, sofosbuvir, GS-441524, and standard N3, is also given in 2D (Figs. S1–S3). The other top-scoring nucleoside precursors and analogs, except sofosbuvir and GS-441524, are methotrexate, pentoxifylline and molnupiravir, exhibits binding with some common residues of the active pocket of FIPV-Mpro i.e., Val26, His41, Thr47, Ala141, Cys144, His162, His163, Leu164, Glu165, Pro188 and Ser189 (Table 3 and Figs. S4–S6). In contrast the, the best docking pose of standard inhibitor (N3) exhibited a binding affinity to Mpro with low binding energy (ΔG= −7.11 Kcal/mol) (Table 3). However, also the highest -CDOCKER energy and -CDOCKER interaction energy score (positive value) referred to the most favorable binding of ligand to the protein. The -CDOCKER energy and -CDOCKER interaction energy score for the N3 best docking pose is 88.72 and 68.11 (Table 3). The interaction between Mpro and N3 involves key amino acid residues and bonds, including Asn25, His41, Ala141, Cys144 Leu164, Glu165, Leu166, Gly167, Pro188, Met190 shown in 2D image (Fig. 2 and Table 3).

Table 3 Summary of the interacting patterns of selected nucleoside precursors and analogs with crystal structure of FIPV-Mpro with ligand N3 (PDBID:5EU8).

S.No	Compounds	-CDOCK Energy	CDOCK Int Energy	Binding energy	MM-GBSA	Ligand interacting residues of Mpro	
1	*N3	88.72	68.11	−7.11	−90.64	Asn25, His41, Ala141, Cys144 Leu164, Glu165, Leu166, Gly167, Pro188, Met190	
2	Sofosbuvir	39.56	39.34	−7.38	−106.82	His41, Cys144, Leu164, Glu165, Leu166, Pro188	
3	GS-441524	41.12	43	−7.2	−91.02	His41, Thr47, His162, His163, Leu164, Glu165, Pro188	
4	Methotrexate	18.71	24.68	−6.98	−88.76	Thr47, Ala141, Leu164, Glu165, Pro188	
5	Pentoxifylline	11.40	39.67	−6.65	−67.42	Thr47, Cys144, His162, Leu164, Pro188, Ser189	
6	Molnupiravir	25.78	25.81	−6.60	−63.24	His163, Leu164, Pro188, Ser189	
7	Tenofovir	−3.03	29.49	−6.4	−62.66	Thr47, His163, Leu164, Glu165, Lgn187, Pro188	
8	6-Azauridine	0.16	26	−6.5	−54.72	Thr47, Cys144, His162, Glu165	
9	Mizoribine	−12.90	18.30	−5.83	−70.64	Ser48, His163, Glu165, Pro188	
10	Oxipurinol	0.45	14	−5.7	−35.16	His41, Thr47, His163, Leu164, Glu165	
11	Galidesivir	−1.87	27.53	−4.59	−46.67	Thr47, Leu164, Glu165, Asp186	
12	Favipiravir	8.77	16.91	−5.32	−45.49	Leu164, Asp186, Pro188	
13	Barcitinib	−29	28	−5.13	−59.06	Cys144, Leu164, Pro188	
14	Ribavirin	−19	22	−5.11	60.24	Thr47, Cys144, His163, Leu164, Pro188	
15	Gemcitabine	12	28	−5.86	−63.37	Thr47, His163, Leu164, Pro188	
Notes.

* The reference standard inhibitor of FIPV-Mpro activity.

CDOCK Int Energy CDOCKER Interaction energy

Figure 2 The best docking hit pose of GS-441524, sofosbuvir, and reference standard (N3) with Mpro.

(A–C) The 3D docking poses of the standard N3 (sea blue), compound Sofosbuvir (green), and GS-441524 interaction with residues of the binding cavity of FIPV-Mpro, respectively. (D) The top hit of GS-441524 (red) and sofosbuvir (green) overlayed on the top pose of reference standard N3 (sea blue) inside the binding cavity of the FIPV-Mpro protein.

Ligand-protein MM-GBSA calculations-pre MD-simulation

The calculation of MM-GBSA for all the compounds were carried out for the top 10 hits and the standard (N3). The top hit of sofosbuvir and GS-441524 displayed the highest free energy (−106.82 kcal/mol and −91.02 kcal/mol), predicting the stability of its complex with the Mpro and validating the docking results. These top hits from sofosbuvir are considered as Ligand 1 and GS-441524 as Ligand-2, compared with the standard N3 (Table 3). The remaining hits displayed comparable free energy when compared to the standard (−90.64 kcal/mol), with only compound oxipurinol exhibiting a significantly lowered free energy of −35.16 kcal/mol (Table 3).

Molecular dynamics simulation

Proteins play a crucial role in various structural and functional processes, including microbial pathogenesis, by mediating receptor-based internalization and replication. Therefore, investigating how ligands such as nucleoside analogs affect the structural stability of FIPV-Mpro is essential to understanding their potential to block viral internalization and replication. Molecular dynamics simulation (MD simulation) is a valuable tool for examining protein structures in silico. In this study, we selected the top-performing nucleoside analogs with the lowest ΔG values from CDOCKER-docking studies such as Sofosbuvir (Ligand 1), GS-441524 (Ligand 2) and N3 (standard inhibitor) for FIPV-Mpro, for 50,000 picosecond (ps) MD simulations. A comprehensive analysis of trajectories, including RMSD, RMSF, RG, number of hydrogen bonds established, and hydrogen bond distance, was measured to explore the stability and molecular interactions of the protein-ligand complexes compared to a standard inhibitor (N3).

RMSD calculations conferring stability of ligand–protein complexes

The RMSD value of < 2.5 Å is considered desirable and indicates similarity to the standard N3-Mpro complex. A lower RMSD value also suggests greater stability of the protein-ligand complexes. In this study, the RMSD of the ligand–protein complexes were significantly less fluctuating for GS-441524 and sofosbuvir as compared N3, during the simulations, indicating high stability of GS-441524 and sofosbuvir with Mpro (Fig. 2). The average RMSDs of the backbone atoms for the GS-441524-Mpro, sofosbuvir-Mpro and N3-Mpro ranged between 1 and 3 nm. Specifically, the average RMSDs of the backbone atoms for the GS-441524–Mpro complexes were 2.2 Å, of two MD runs respectively. The complex of ligand 2- GS-441524 showed stability after the initial 0–12,000 ps, while it showed slight fluctuation after the 8,000–11,000 ps (Fig. 3) with the FIPV-Mpro during MD, which quickly stabilized again to the average RMSD (2.2 Å). The complex of ligand 1-sofosbuvir-Mpro showed stability after the initial 0–17,000 ps, while it showed slight fluctuation after the 17,000–21,000 ps (Fig. 3) with the FIPV-Mpro during MD, which quickly stabilized again to the average RMSD (2.8 Å). Conversely, the standard compound- N3-Mpro complex also showed an average RMSD of 2.7 Å, MD runs, respectively.

Figure 3 Results of the analysis of the RMSD values of the backbone atoms for the standard inhibitor (Std) N3-Mpro and the various protein–ligand complexes.

The RMSD profile values for the backbone atoms of the standard inhibitor standard (Std) N3-Mpro and protein-ligand complexes of sofosbuvir as ligand-1 (Lig-1)-Mpro and GS-441524 ligand-2 (Lig-2) -Mpro from initial structures to complete MD simulation period. The RMSD values are high for both runs of Mpro-N3 (std) as compared to Mpro-Lig-1 and Mpro-Lig-2. The lower the RMSD, the greater the stability of the protein complex.

Stability and mobility of complexes-RMSF

The predicted RMSF graphs of the C-α atom of all five complexes were plotted against the interacted residues based on the trajectory period of the MD simulation. All the residues in docking complexes have fluctuated around 0.5–3 Å in the simulation time scale. The RMSF values for the amino acids in the sofosbuvir as Ligand-1-Mpro complex were notably lower than those in the N3-Mpro complexes, while the compound GS441524 as Ligand-2-Mpro complexes suggesting near equal residue mobility Mpro protein with N3-inhibitor complex (Fig. 4). The significance of the low RMSF values from the RMSF analysis being an essential tool in the identification of the rigid and flexible sections of the protein structure. This conclusion is further validated after considering their low RG values, which indicated the low flexibility of both standard and inhibitor complexes.

Figure 4 RMSF analysis of mpro complexes with standard inhibitor N3, sofosbuvir (Lig-1), and GS-441524 (Lig-2) during 50,000 ps simulation.

The RMSF of the standard (N3)-Mpro, Sofosbuvir (Lig-1)-Mpro, and GS-441524 (Lig-2)-Mpro complexes during 50,000 ps simulation. The RMSF of alpha carbons shows no fluctuations of a loop region for both runs of Mpro-N3 (std) as compared to Mpro-Lig-1 and Mpro-Lig-2. The RMSF plot suggested that equal and lower the RMSF value compared to the standard N-Mpro suggested stability of the Lig-1 (sofosbuvir) and Lig-2 (GS-441524) with Mpro protein complex.

Complex compactness analysis-radius of gyration (RG)

RG is used to determine whether the complexes are stably folded or unfolded during the MD simulation. The average RG value of two MD runs of N3-Mpro was calculated to be around 22.2 Å. Furthermore, the average RG values of sofosbuvir (Ligand-1) with Mpro complex and GS-441524 (Ligand 2) with Mpro were 21.9 Å and respectively, significantly similar to reference N3-Mpro complex which is 22.2 Å (Fig. 5). As a result, it can be observed, that GS-441524-Mpro and sofosbuvir-Mpro complex exhibited relatively similar behavior of compactness and consistent values of RG as compared to the reference standard N3-Mpro. It indicates that these are perfectively superimposed with each other and have high stability.

Figure 5 Radius of gyration analysis of Mpro complexes with standard inhibitor N3, sofosbuvir (Ligand-1), and GS-441524 (Ligand-2) over 50,000 ps simulation.

The radius of gyration of the standard (N3)-Mpro, sofosbuvir (Ligand-1)-Mpro, and GS-441524 (Ligand-2)-Mpro complexes during 50,000 ps simulation. Plot of RG reflecting the changes observed in the conformational behavior of the all protein-ligand complexes compared with standard complex (N-Mpro).

Hydrogen bonds and its distance-Complex stability

Molecular dynamics simulation had been carried out through 50,000 ps simulation time for all new compounds against Mpro protein. To investigate the stability of the ligand interaction with protein, the measurement of intermolecular hydrogen bond development and its distance between ligand-protein complex were evaluated. The complexes of sofosbuvir (Ligand-1) and N3 (Standard) with Mpro protein protease maintained two hydrogen bonds throughout the entire simulation time. However, the GS-441524 (Ligand-2) was able to maintain three hydrogen bonds throughout the simulation time confers the stability of ligand interaction with Mpro protein (Figs. 6B and 6D). Furthermore, all the three complexes in two runs of MD were able to maintain four hydrogen bonds through most of the conformations of simulation (Figs. 6A and 6F).

Figure 6 Hydrogen bond analysis between Mpro and ligands (Sofosbuvir, GS-441524, and N3) during 50,000 ps MD simulations.

Hydrogen bond (Hbond) monitor plots showed the number of hydrogen bonds established during 50,000 picosecond simulation time. (A and C) A hydrogen bond occurs between Mpro-protein and Ligand-1 (sofosbuvir) in the first and second run of MD. (B and D) A hydrogen bond occurs between Mpro-protein and Ligand-2 (GS-441524) in the first and second run of MD. (E and F) Hydrogen bond occurs between Mpro-protein and standard (N3) in the first and second run of MD.

On the other hand, the estimation of hydrogen bond distance monitor between the ligands and the interacted residues of the Mpro-protein, the sofosbuvir (Ligand 1) showed four hydrogen bonding interactions during simulation time with the Mpro protein. In the depicted hydrogen bond distance graph of two MD simulation runs for sofosbuvir, residue Cys144, Gly167, THR and Glu165 interaction shown weak hydrogen bonding, on the basis of distance of hydrogen bond interaction of Ligand-1 (sofosbuvir) with Mpro. Mpro Cys144 residue showed average bond distance at 2.9 Å (Fig. 7A). All four bonds shown average bond distance fluctuation 1.9–8.9 Å, during the whole simulation time. While in the second run of MD, the interaction of Ligand 1 and Cys144, Gly167 and Glu165 appeared at about average bond distance of 2.9 Å, 3.1 Å and 3.2 Å and average bond distance fluctuation is 1.9–8 Å, shown less stable interaction between sofosbuvir with Mpro (Fig. 7B).

Figure 7 Hydrogen bond distance analysis between sofosbuvir (Ligand-1) and Mpro binding-site residues during MD simulations.

Hydrogen bond distance plot of compound sofosbuvir (Ligand-1) with Mpro protein. (A and B) The plot indicates changes in distance between key binding-site amino acids and the ligands over time during each run of MD simulation.

In contrast, GS-441524 (Ligand 2) exhibited stable binding with Mpro, showed interaction of Glu165 with average hydrogen bond distance of 2.2 Å throughout the simulation time. The other residues like Ser189, Gly167, and Pro188 showed an average hydrogen bond distance between 2.9 Å, while average bond distance fluctuation is 1.9−4.7 Å, conferring stable interaction between GS-441524 and Mpro (Figs. 8A and 8B). The standard reference ligand, N3 compound against Mpro protein during simulation time, showed interaction with Glu165 and Ser189 with an average bond distance of 2.5 Å, conferring stable interaction between standard N3 and Mpro. However, the residue Pro188 and Asn25 also exhibited an average hydrogen bond distance of 2.8 Å (Figs. 9A and 9B).

Figure 8 Hydrogen bond distance plots of GS-441524 (Ligand-2) interacting with Mpro protein.

Hydrogen bond distance plot of compound GS-441524 (Ligand-2) with Mpro protein. (A and B) Plot indicates changes in distance between key binding-site residues and the ligands over time during each run of MD simulation.

Figure 9 Hydrogen bond distance analysis between standard inhibitor N3 and Mpro binding-site residues during MD simulations.

The hydrogen bond distance plot of compound N3 (standard) with Mpro protein. (A and B) Plot indicates changes in distance between key binding-site amino acids and the ligands over time during each run of MD simulation.

Discussion

Feline infectious peritonitis (FIP) is a severe disease in cats caused by the feline infectious peritonitis virus (FIPV), a variant of the feline coronavirus (FCoV). Normally causing mild enteritis, FCoV transforms into FIPV, affecting multiple organs in cats (Sherding, 2006). FIPV is an enveloped, positive-sense, single-stranded RNA virus classified within the Coronaviridae family, which also includes viruses like SARS-CoV, MERS-CoV, and SARS-CoV-2. Among humans and animals, coronaviruses exhibit the presence of similar structurally related functional proteins, such as the Mpro, also known as 3CL protease (3CLpro), which plays a crucial role in viral replication (Theerawatanasirikul et al., 2020). In this study, the initial phases of a drug screening for Mpro inhibitors, imply and involve drug-likeness assessment of different chemical compounds with the help of various factors. In this context, various computer-aided approaches, such as Lipinski’s Rule of Five, are commonly applied (Ahmad et al., 2024; Jia et al., 2020).

The LogP values of all the compounds provided in Table 1, suggested that these compounds are likely to be more hydrophilic and may have better solubility in water, which can be beneficial for certain therapeutic applications where high aqueous solubility is required. This low LogP also indicates favorable hydrophobicity, contributing to increased persistence and bioavailability by reducing renal excretion (Alvi et al., 2017).

In drug discovery, an MPSA value of ≤ 140 Å2 is often considered an optimal threshold for oral bioavailability (Veber et al., 2002). Drugs with higher MPSA values are less lipid-soluble and distributed less extensively attributed to its less extensive and slow absorption rate, than drugs with lower TPSA values. All the nucleoside analogs and precursors showed MPSA values below the threshold. As a result, these compounds exhibit enhanced bioavailability and improved pharmacological efficiency. Based on these findings, all the selected nucleoside precursors and analogs qualifying the criteria for MPSA. Compounds showing slight increased MPSA values (near threshold ≤ 140 Å2) can be considered for further drug screening evaluations.

In ADMET analysis, aqueous solubility refers to the ability of a compound to dissolve in water, which is crucial for the compound’s absorption, bioavailability, and pharmacokinetic properties. The aqueous solubility of a drug influences how well it can be absorbed in the gastrointestinal tract and how effectively it can be distributed throughout the body. However, the compound mizoribine, 6-Azauridine, and ribavirin shows solubility level 5 (Table 1), indicates compounds are extremely soluble, conferring unsuitable for drug likeliness of these compounds on the basis of solubility parameter of ADMET. On the other hand, the blood brain barrier (BBB) model predicts blood–brain penetration after oral administration. In our investigation, all 14 selected compounds showed low penetration values for BBB. Plasma protein binding of drug molecules can affect the efficiency of a drug, because the bound fraction is temporarily shielded from metabolism. On the other hand, only the unbound fraction exhibits pharmacological effects. Drugs with extreme PPB reflect a low volume of distribution (Vd), long plasma half-lives (T1/2), and may incur lower hepatic and renal clearance (Gurevich, 2013; Roberts, Pea & Lipman, 2013).

On the basis of ADMET model predicted PPB, the model predicted false PPB for all the compounds, suggests that, according to the model, these compounds are not predicted to have significant plasma protein binding (Table 2). This implies that the compounds do not strongly interact with plasma proteins, remain largely in their free form, which could lead to greater bioavailability, easily distribute to tissues and reach its site of action and enhanced therapeutic effects. From intestinal absorption analysis prediction results, compounds with intestinal absorption levels of 0–2 indicate good to moderate absorption and enhanced bioavailability, linked to the pharmacological actions of nucleoside analogs (Table 2). In contrast, inhibitors like N3, methotrexate, mizoribine, ribavirin, and 6-Azauridine, with an absorption level of 3, suggest poor intestinal absorption. However, a compound with less absorption is not directly correlated with its pharmacological potential.

Based on the CYP2D6 values and corresponding predictions, out of 14 compounds, none of these drugs are expected to strongly interact with the CYP2D6 enzyme. Similarly, Mpro-standard inhibitor N3 also did not inhibit the CYP2D6 activity. Therefore, they are likely neither inhibitors nor substrates for CYP2D6, which may reduce the risk of metabolic interference. These results imply that the selected nucleoside analogs are likely to undergo rapid metabolism and excretion, potentially contributing to their potent pharmacological effects and reduced toxicity. CYP2D6 is involved in the metabolism of a wide range of substrates in the liver, and its inhibition by a drug constitutes a majority of cases of drug-drug interaction, mediates around 25% of total metabolism and clearance of the drugs after administration (Ahmad et al., 2024). Furthermore, the toxicity assessment of nucleoside precursor and analogs results concluded that Ames test predicts 6-Azauridine, galidesivir, and tenofovir as mutagenic, while the remaining compounds are non-mutagenic. Additionally, the negative hepatotoxicity and Ame’s test values suggest that most compounds are less toxic and safe from mutagenic effects (Table 2). In conclusion, the drug likeliness and ADMET properties prediction results suggested that the selected nucleoside precursors and analogs may exhibits instant metabolism and excretion which might be attributed to their substantial pharmacological effects as well as low toxicity.

Further, the molecular docking analysis of all selected compounds was conducted to identify potential anti-FIPV agents from a pool of nucleoside precursors and analogs. Using CDOCKER for molecular docking, we evaluated and ranked the binding affinities of promising compounds with the catalytic dyad residues (His41 and Cys144) in the binding pocket of FIPV-Mpro. The binding affinity (-CDOCKER interaction energy) scores of the all the compounds with Mpro were ranged from 68.11 to 28 (Table 3). These compounds were further selected to calculate ΔG through calculate binding energy method. The binding energy scores of all interactions between ligands and the Mpro were ranged between −7.11 to −4.59 kcal/mol (Table 3). Out of 14 selected compounds, top two compounds which exhibited better binding affinity in catalytic dyad pocket, are sofosbuvir (Ligand-1) and GS-441524 (Ligand-2) (Table 3). The binding score and energy of all the compounds in including reference ligand (N3) with FIPV-Mpro is shown in Table 3. The interaction of N3 establishes interaction with Mpro through binding with specific residues Cys-144 and His-41 of catalytic dyad of Mpro protein. Most interestingly, the residues stabilizing the binding of top scoring nucleoside analog (sofosbuvir and GS-441524) against FIPV-Mpro also exhibited interaction with residues of catalytic dyad. Therefore, due to high negative binding energy and interaction pattern, GS-441524 and sofosbuvir were slightly different than that of other top-scoring nucleoside analogs and precursors (methotrexate, pentoxifylline and molnupiravir) suggesting that these potent inhibitors of FIPV-Mpro can be used synergistically to produce enhanced synergistic protective effects against FIPV infection via restricting the proteolysis of polyproteins and thereby restricting the expression of FIPV viral proteins/enzymes. The catalytic dyad residues Cys-144 and His-41, plays crucial role in Mpro enzymatic activity, facilitating the cleavage of viral polyproteins into functional proteins essential for viral replication (Jin et al., 2020). This catalytic dyad is pivotal for the enzymatic activity of Mpro, facilitating the cleavage of viral polyproteins into functional proteins essential for viral replication and assembly (Galasiti Kankanamalage et al., 2018; Jin et al., 2020). After takeover of the host’s transcriptional machinery, coronaviruses makes host cell synthesize two overlapping polyproteins, pp1a and pp1ab, which cleaves through coronavirus-encoded proteases—papain-like protease (PLpro) and Mpro. The cleavage of pp1a and pp1ab forms 16 nonstructural proteins play roles in viral replication (Lu et al., 2022). Owing to the determining role in the proteolysis of viral polyproteins, the Mpro of the FIPV has been established as the preferred target in combating its virulence. Similarly, we also used FIPV-Mpro as the major target via selected nucleoside analogs for the management of this highly lethal infectious virus of domestic cats.

For the analysis of the stable interaction of ligands with Mpro docking complexes were further investigated through MD simulation parameters. The RMSD was analyzed to assess the variations in the backbone Cα atoms of target proteins resulting from the binding of top-ranking nucleoside precursors and analogs along with their respective reference standards (N3). Ideally, RMSD values would be zero; however, due to statistical uncertainties, it is not possible for a protein to have an RMSD of zero (Ahmad et al., 2024). The RMSD plots of each complex clearly demonstrate that the all compounds–Mpro complexes showing low RMSD values (RMSD < 2.5 Å), confirming the stability of complexes during the simulation period when compared to the standard N3-Mpro complex (Fig. 3). RMSD played a significant role in protein stability (Elengoe, Naser & Hamdan, 2014).

Unlike RMSD, the residual fluctuations in the MD simulation complexes were observed by RMSF (Ashraf et al., 2016). The RMSF is also defined as the standard measure of deviation of a molecule from its initial position (Swetha, Ramaiah & Anbarasu, 2016). The RMSF quantifies the overall structural deviations over time and measures the root mean square fluctuations of individual amino acid residues, providing insight into the dynamic behavior and flexibility of these residues during simulations. This analysis enables the identification of critical residues contributing to conformational flexibility within the protein. In this study, a comprehensive RMSF analysis of the top ligand-protein complexes was performed to evaluate the mobility of residues within the protein’s active site. The fluctuations observed in the RMSF profiles throughout the simulation runs pinpointed regions of the protein undergoing significant motion. Investigation of RMSF showed low or equal RMSF for all the complexes as compared to standard N3-Mpro complex (Fig. 4). In conclusion, it indicated that RMSF of all complexes is significantly similar compared to reference standrad N3-mpro complex residues, exhibiting less fluctuation which is substantial enough to be considered as good stability of complexes (Mathpal et al., 2022).

However, the compactness of the target protein-ligand complex was measured through RG (Abbas et al., 2017). The stability of the FIPV- Mpro with ligands exhibiting the best binding energies was further evaluated using the RG, which measures the protein’s size and compactness (Arnittali, Rissanou & Harmandaris, 2019). The magnitude of RG inversely correlates with protein stability, where a larger RG indicates a less stable, more expanded structure. The RG value GS-441524 -Mpro complex was 0.2−0.4 Å low comparable to that of the N3-Mpro complex. However, the RG value of the sofosbuvir-Mpro complex was similar to N3-Mpro complex over the 50,000 ps simulation, suggesting that the protein is less compact and denser upon complex formation (Fig. 5). In conclusion, the higher RG value attributed to low compactness of ligand-protein complex. This indicates that GS-441524-Mpro and sofosbuvir-Mpro complexes demonstrated significant structural compactness and stability. If the protein likely to maintain a relatively steady value of RG over the MD simulation, it is regarded as s folded and more stable, and if its RG changed over time, it would be considered unfolded (Ghasemi et al., 2016).

Further, the number of hydrogen bond establishments and changes in the distance of the compounds to the amino acid residues contributing to substantial hydrogen bonds were assessed by studying the variation of distance between Ca atoms of the binding cavity and the compounds throughout the MD simulations. Based on the simulation result, the average number of hydrogen bond establishment and hydrogen bond distance between the compounds and the contributing residues was shown by GS-441524-Mpro as compared to standard N3-Mpro, which is below 2.5 Å. During the MD simulations of the compound sofosbuvir (Ligand-1) and the distance between the compound and Mpro, both Ser189 and Val42 amino acid residues showed high fluctuations compared to Standard N3 and GS-441524. While, compound GS-441524 (Ligand-2) and Mpro interacted residues showed less fluctuation for hydrogen bond distance (Fig. 8), as compared to standard N3-Mpro and (Fig. 9) and Ligand-1 sofosbuvir-Mpro interacted residues (Fig. 7). More the fluctuation in hydrogen bond distance attributed to the less effective binding of ligand with residues of active pocket of protein (Rasyid, Purwono & Pranowo, 2020). While, the average distance between the compounds and the hydrogen bond contributing residues should be near or below 2.5 Å which falls within the accepted range (Nada, Elkamhawy & Lee, 2022). The residues stabilizing the binding of compounds exhibiting low hydrogen bond distance particulary the GS-441524 and sofosbuvir against FIPV-Mpro were different than that of top-scoring residues of reference standard N3, suggesting that these potent inhibitors can be used synergistically to produce enhanced synergistic protective effects against FIPV infection via restricting the proteolysis of polyproteins and related expression of viral proteins. Restricting the expression of viral proteins/enzymes, as a combinatorial effect of compounds is reported for SARS-CoV-2 inhibition (Keretsu, Bhujbal & Cho, 2020; Ullrich & Nitsche, 2020). Therefore, the results of findings such as docking analysis of different MD simulation parameters including hydrogen bond distance analysis compared with N-Mpro complex, validate the GS-441524 as FIPV-Mpro inhibitor.

Gilead Sciences (GS)-441524 is a 10-cyano adenosine analog, which is the main plasma metabolite of the more famous antiviral drug remdesivir (Amirian & Levy, 2020). Several cellular studies conducted on GS-441524 indicated an anti-SARS-CoV-2 activity comparable when not higher than remdesivir (Yan & Muller, 2020), with some studies pointing out that GS-441524 would be even more convenient than remdesivir for the COVID-19 therapy (Yan & Muller, 2020), GS-441524 advantages over remdesivir include ease of synthetic preparation, lower hepatic toxicity, as well as oral administration route (not suitable for remdesivir due to its poor liver stability) (Yan & Muller, 2020). However, Sofosbuvir is also a nucleoside analog used to treat HCV infection, which can inhibit the SARS-CoV-1 RNA-dependent RNA polymerase, affecting the viral life cycle (Jacome et al., 2020). Sofosbuvir-based treatment regimens may help to reduce the mortality of patients with SARS-CoV-2 and improve associated complications (Hsu et al., 2022). The Sofosbuvir is a prodrug that is hydrolyzed by liver enzymes after absorption to form the monophosphate uridine analog, which is further phosphorylated to form the active triphosphate form (Alrehaily et al., 2023).

In our docking study, GS-441524 interaction could inhibit its binding with polyproteins, pp1a and pp1ab. Our docking and molecular dynamic simulation study parameters confirm that developing a single antiviral agent targeting Mpro, suggesting GS-441524 is either used as a drug alone in treatment or in combination with other potential therapies. This could serve as a potential system that could pave the way to defense against diseases associated with animal coronaviruses. Additionally, our findings provide valuable insights for advancing the development of antiviral agents and broaden the range of anti-CoV agents available for combating feline coronavirus and other related coronaviruses.

The current study shows the promising results of using a newly designed drug targeting the FIPV-Mpro (GS-441524) as well as the repurposing of the Sofosbuvir (FDA-approved drug for the hepatitis-C virus (Summers, 2018). This study supports the strategy of the design and repurposing of some antiviral drugs for one of the most serious viral diseases of cats that cause high mortality rates among the affected cats. The outcomes of this strategy will improve the health care of cats and improve FIPV management in crowded cat populations as shelters, as well as increase the cat survival rate due to FIPV infection in the future. This approach will be an ideal example of the success of the design/repurposing of some drugs for some viral diseases in cats.

The in silico findings presented here offer valuable insights into the potential activity of the studied compounds, the absence of experimental validation represents a key limitation. The computational predictions, while useful for hypothesis generation, must be supported by in vitro and in vivo studies to confirm their biological and therapeutic relevance. Future work will focus on experimentally validating these predictions to establish their translational significance.

Conclusions

The results from this study support the development of a single antiviral agent GS-441524, Mpro targeting agent, and its combination with other potential therapies could provide an effective first line of defense against diseases associated with feline coronaviruses. Nevertheless, this is an initial in-silico study and should be validated through in-vitro and in-vivo experiments in infection models. Additionally, the predicted binding affinity profiles, stability analysis ADME, and toxicity of the selected nucleoside precursors and analogs based on algorithm-based tools and approaches may vary in experimental settings. While the in-silico findings are promising, experimental validation through in vitro and in vivo studies remains essential to confirm their biological and therapeutic relevance. Further, our findings offer essential insights for the further development of antiviral agents and expand the reservoir of anti-FIPV agents for targeting feline coronavirus and other related.

Supplemental Information

Supplemental Information 1 Supplementary tables and figures

Additional Information and Declarations

Competing Interests

Author Contributions

Data Availability

1 Portions of this text were previously published as part of a preprint (Khan et al., 2024).

The authors declare there are no competing interests.

Mohd Yasir Khan performed the experiments, analyzed the data, prepared figures and/or tables, authored or reviewed drafts of the article, and approved the final draft.

Abid Ullah Shah performed the experiments, analyzed the data, prepared figures and/or tables, authored or reviewed drafts of the article, and approved the final draft.

Nithyadevi Duraisamy performed the experiments, analyzed the data, prepared figures and/or tables, authored or reviewed drafts of the article, and approved the final draft.

Nadine Moawad performed the experiments, authored or reviewed drafts of the article, and approved the final draft.

Reda Nacif ElAlaoui conceived and designed the experiments, performed the experiments, analyzed the data, authored or reviewed drafts of the article, and approved the final draft.

Mohammed Cherkaoui conceived and designed the experiments, performed the experiments, analyzed the data, authored or reviewed drafts of the article, and approved the final draft.

Maged Gomaa Hemida conceived and designed the experiments, performed the experiments, authored or reviewed drafts of the article, and approved the final draft.

The following information was supplied regarding data availability:

The data is available at figshare: Khan, Mohd Yasir; Hemida, Maged Gomaa (2025). Supplemental Information. figshare. Dataset. https://doi.org/10.6084/m9.figshare.29298449.v1.

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
