# Peer review of "Identification of potential inhibitors of the main protease from feline infectious peritonitis virus using molecular docking and dynamic simulation approaches"

_PeerJ, doi:10.7717/peerj.19744_

## Round 0.1 · original submission · Major Revisions

Please address all reviewers comments.

**Language Note:** The review process has identified that the English language must be improved. PeerJ can provide language editing services - please contact us at [email protected] for pricing (be sure to provide your manuscript number and title). Alternatively, you should make your own arrangements to improve the language quality and provide details in your response letter. – PeerJ Staff

·

Basic reporting

The introduction section includes detailed virological and morphological characteristics of the virus; however, the overall flow could benefit from clearer transitions between paragraphs. It would be more effective to provide background information on the clinical manifestations of the virus in cats, the challenges associated with its treatment, the populations most in need of antiviral therapy, and diagnostic approaches used to identify these cases. Including this information would better support the rationale and objectives of the study. Additionally, the hypothesis should be more clearly stated alongside the aim of the study.

It is also recommended that transitions between paragraphs be reviewed in terms of fluency and rationality.

Experimental design

Since this section contains too many specific and technical details, some parts need to be clarified (selection criteria, method preference, etc.).

Validity of the findings

The findings are explained comprehensively. However, additional comments are provided in the file and below.

Additional comments

The discussion section is detailed, and the findings are well-explained. I believe this study has the potential to contribute meaningfully to the development of practical treatment protocols that can be applied in clinical practice. However, due to the study’s highly specific and technical nature, it may be challenging for general veterinary clinicians to fully grasp its clinical relevance. Therefore, adding a paragraph that highlights its clinical significance would enhance the applicability and impact of the manuscript. Additionally, considering that certain technical aspects fall outside my area of expertise, I recommend that the materials and methods section, particularly the technical procedures, be reviewed by referees with expertise in the relevant methodologies to ensure accuracy and completeness.

Thank you.

Reviewer 2 ·

Basic reporting

Some minor typographical issues and grammatical inconsistencies

The introduction could benefit from a clearer statement of the novelty of this study in relation to prior FIPV docking studies.

The figure legends need to be more self-explanatory; for instance, figures showing RMSD/RMSF should include interpretations of what specific fluctuations imply.

Experimental design

Clearly define what is meant by “artificial intelligence” in the title—does it refer to specific ML tools or predictive algorithms?

Consider including redocking validation or RMSD between predicted and experimental N3 pose to demonstrate the reliability of the docking protocol.

Validity of the findings

There is no negative control or decoy compound to assess specificity.

All results are in silico; the conclusion calls for in vitro/in vivo validation, but this could be further emphasized as a limitation.

Additional comments

None

---

## Round 0.2 · accepted · Accept

Thanks for addressing all comments.

·

Basic reporting

In line with previous evaluations, the authors have corrected the relevant parts. They also appear to have checked the entire text for grammar and spelling rules.

Experimental design

The material and methods section has been revised and made clearer.

Validity of the findings

The findings are stated clearly.

Additional comments

I believe that the revised version is sufficient. Thank you.